# Longitudinal Associations of Dietary Fructose, Sodium, and Potassium and Psychological Stress with Vascular Aging Index and Incident Cardiovascular Disease in the CARDIA Cohort

**DOI:** 10.3390/nu16010127

**Published:** 2023-12-29

**Authors:** Meaghan Osborne, Alexa Bernard, Emily Falkowski, Deni Peterson, Anusha Vavilikolanu, Dragana Komnenov

**Affiliations:** 1Nephrology and Hypertension, Internal Medicine and Physiology, School of Medicine, Wayne State University, Detroit, MI 48201, USA; meaghan.osborne@med.wayne.edu (M.O.); alexa.bernard@med.wayne.edu (A.B.); hl4960@wayne.edu (E.F.); hk8129@wayne.edu (D.P.); avavili@wayne.edu (A.V.); 2John D. Dingell VA Medical Center, Detroit, MI 48201, USA

**Keywords:** vascular aging, cardiovascular disease, lifestyle

## Abstract

We explored how dietary behaviors (sucrose, fructose, sodium, and potassium consumption) and endured psychological stress in young adult males and females impact the vascular aging index (VAI) and CVD risk by mid-life. Data were obtained from the Coronary Artery Risk Development in Young Adults Study, an ongoing longitudinal study. The included participants (n = 2656) had undergone carotid artery ultrasound at year 20 allowing VAIs to be calculated. Demographics, dietary data, and depression scores were obtained at baseline and year 20 of follow-up. Regression analyses were used to assess the predictors of VAI. Cox regression analyses were conducted to assess the risk of CVD, stroke, and all-cause mortality. Predictors of vascular aging were found to be sex-specific. In females, depression scores at baseline were positively associated with VAI (B-weight = 0.063, *p* = 0.015). In males, sodium intake at year 20 positively predicted VAI (B-weight = 0.145, *p* = 0.003) and potassium intake inversely predicted VAI (B-weight = −0.160, *p* < 0.001). BMI significantly predicted CVD, stroke, and death. Fructose consumption at year 20 was a significant predictor of CVD risk while having high blood pressure at baseline was significantly associated with stroke risk. Our findings support the promotion of nutrient-specific behavior changes to prevent vascular aging in early adulthood and CVD risk in mid-life.

## 1. Introduction

Cardiovascular diseases (CVDs) are the leading cause of death and disability worldwide [1]. CVD risk increases exponentially with age, largely due to pathophysiological changes in the vasculature, namely calcification, loss of elastin, increased collagen deposition, and increased vascular diameter [2]. The major consequence of vascular aging is increased arterial stiffness, which decreases compliance and the ability of the vessel to adapt to stressors [2]. The gold standard measurement of arterial stiffness is carotid–femoral pulse wave velocity (cfPWV) which has been shown to be a strong predictor of future CVD events [3]. Recent data suggest that the combination of the vessel structural properties (i.e., carotid intima–media thickness (cIMT)) with the functional properties (i.e., PWV) into a vascular aging index (VAI) may improve the prediction of CVD risk [4]. Many mechanisms have been proposed to explain the vascular aging process including oxidative stress, chronic inflammation, and the cellular mechanisms driven by the mammalian target of rapamycin signaling pathway [5]. In addition, it is well known that hypertension leads to early vascular aging [6], as well as certain lifestyle factors including psychological stress and diet [7,8,9]. Given that the U.S. population over 65 years is expected to nearly double by 2060, and that age is a major risk factor for CVDs, there is a need to elucidate the mechanisms that affect the vascular aging process [10].

According to the 2018 National Health Interview Survey, males have a higher prevalence of heart disease, coronary heart disease, hypertension, and stroke (12.6%, 7.4%, 26.1%, and 3.1%) when compared to age-matched females (10.1%, 4.1%, 23.5%, and 2.6%) [1]. Sex hormones, mainly estrogens, have been used to explain such differences, given that the post-menopausal period is associated with an increased CVD prevalence and heart failure development in females [11]. One of the mechanisms proposed includes estrogen-mediated β-adrenergic vasodilation [12]. The hypothesis that estrogen is cardioprotective is supported by the finding that pre-menopausal arteries have significantly less endothelial inflammation in the setting of enhanced vasodilatory capacities [13]. This, in turn, prevents oxidative damage, which is a significant mediator of arterial stiffening and thus vascular aging [14]. Any protective effects of estrogen pre-menopause appear to be eliminated post-menopause whereby post-menopausal females experience a rapid increase in blood pressure, calcification scores, and central arteriole stiffness [14].

In addition to intrinsic phenotypic sex differences, it is now recognized that extrinsic or psychosocial factors also contribute to the sex differences in CVD risk and vascular aging [14]. For example, females are at a two-fold greater risk of experiencing depression compared to males and depression is a known CVD risk factor [15]. Adolescent females may experience more interpersonal stress [16]. In addition, childcare responsibilities may impact the prevalence of depression and anxiety in females more than males [17]. The effects of depression on vascular aging have been largely unexplored. Given the sex and gender differences in CVD risk factors, CVD outcomes, and vascular aging, it has been suggested that future studies should incorporate sex disaggregated data for traditional and non-traditional risk factors to help inform precision medicine [18].

Diet constitutes a modifiable lifestyle factor with potential consequences for vascular aging. According to the AHA’s Life’s Essential 8, a diet rich in fruit and vegetables, complex carbohydrates, fish, beans, fiber, moderate in meat, and minimal in processed sugary foods can prevent endothelial dysfunction and arterial stiffness [7,19]. While these recommendations offer broad, healthy dietary patterns, there has been an impetus for studies focusing on specific micro- and macronutrients within those patterns for promoting healthy vascular aging [20]. One relatively well-studied dietary component is sodium. Reducing sodium intake has been shown to consistently lower arterial stiffness in randomized, controlled and crossover trials with normotensive and hypertensive middle-to-older-aged males and females [21]. The limitations of these clinical trials include the small sample sizes and limited racial and ethnic diversity [22]. Moreover, a recent trial in individuals with heart failure (HF) showed that reducing dietary sodium did not improve clinical events compared to those HF patients who did not reduce dietary sodium [23]. Indeed, additional epidemiological data are needed to explore the associations between dietary sodium and vascular aging in the context of covariates important for CVD development: sex, race and ethnicity, hypertension status, and other dietary culprits, such as high fructose corn syrup (HFCS). Given that dietary potassium may also play a vital role in modulating CVD risk, exploring the effects of potassium intake on vascular aging is also warranted [24,25].

Dietary sugar is a disaccharide of glucose and fructose, but an important distinction exists in the fructose component in North American countries compared to some of the European countries (i.e., Scandinavia). Where sugar-sweetened processed foods contain HFCS, the fructose component is typically in excess of glucose by about 50% [26,27]. In fact, since the advent of HFCS in the 1970s, this has been the primary component of sugar used by the food industry in North America [28]. Indeed, pre-clinical work showed that even a short-term moderate increase in dietary fructose causes salt-sensitive blood pressure, diastolic dysfunction, and increased aortic PWV in rats [29,30,31,32]. Additionally, other rodent models showed that the increased ingestion of fructose and salt during early age (i.e., equivalent to adolescence in humans), contributes to hypertension, vascular stiffness, and a decrease in glomerular filtration rate (GFR) later in life even after the exposure to fructose and salt had been removed and later re-introduced to rats once they were older (i.e., equivalent to mid-life in humans) [33,34]. Likewise, in a systematic review and meta-analysis of six human cohort studies, sugar-sweetened beverages were significantly associated with an increased risk of hypertension, although vascular aging variables were not measured [35].

In the present investigation, we included the participants from the CARDIA (Coronary Artery Risk Development in Young Adults) study. To our knowledge, few studies have explored the role of diet on vascular aging within the CARDIA cohort. Gao et al. [36] found that among CARDIA participants, carbohydrate intake was negatively correlated with the risk of coronary artery calcium progression, a marker of atherosclerosis. Duffey et al. [37] showed that sugar-sweetened beverage consumption was positively associated with waist circumference, triglycerides, LDL cholesterol, and hypertension. The objective of the present retrospective observational study was to assess how dietary behaviors, specifically intake of fructose and sodium, and endured psychological stress in young adult males and females impact the VAI and CVD risk by mid-life.

## 2. Materials and Methods

### 2.1. Study Sample

This is a retrospective observational study in which the study sample included the participants from the CARDIA (Coronary Artery Risk Development in Young Adults) cohort designed to examine lifestyle and behavioral factors that contribute to the development of CVDs from young adulthood through midlife. The study recruited 5114 individuals, aged between 18 and 30 years (median = 26 years; this constitutes the baseline time point) in 1985 and 1986 at four centers across the United States: Birmingham (AL), Chicago (IL), Minneapolis (MN), and Oakland (CA), matched for sex, age, race (Blacks and Whites) and education level. Assessments occurred at baseline (year 0, median age = 26) and at years 2, 5, 7, 10, 15, 20, 25, 30, and 35 (ten total assessments to date). The inclusion criteria of the CARDIA study were aged between 18 and 30, Black and White men and women. Other details on the study were published elsewhere [38]. Our inclusion criteria consisted of having completed the carotid ultrasound Doppler scans at year 20 of follow-up. Here, we considered the records from n = 2656 participants for which carotid ultrasound scans were completed and carotid PWV obtained during visit 7 (year 20 of follow-up). The baseline values from visit 1 and outcomes (CVDs, stroke, and death) from the latest visit (follow-up year 35) were obtained only for these individuals (n = 2565) rather than an entire starting cohort of 5114 people. Therefore, no missing data were observed for the cPWV and cIMT which were used in the derivation of VAI. Overall, missing data were less than 5% for the remaining variables, and they were not imputed. The study was approved by the Institutional Review Board at Wayne State University (IRB # 063319MP2X). Most recent approval of the Research Materials Distribution Agreement by the National Institutes of Health, Heart, Lung and Blood Institute was completed on 15 April 2022.

### 2.2. Study Measures and Outcomes

VAI was obtained as described previously [4], with modification of using carotid PWV (cPWV) instead of aortic PWV (aPWV). The value for cPWV was calculated by averaging the PWV values obtained on the left- and right-side vessels expressed in the units of m/s. Carotid IMT (cIMT) was expressed in mm. The modified equation used to calculate VAI was:VAI = (log (1.09) × 10cIMT + log (1.14) cPWV) × 39.1 + 4.76.(1)

Blood pressures were measured in triplicate with a random zero mercury sphygmomanometer for visit 1 by centrally trained personnel. For visit 7, an Omron HEM907XL oscillometric monitor (Omron Healthcare, Lake Forest, IL, USA) was used for obtaining BP values in triplicate after a 5 min rest period using the participants’ right arm. The inter-device differences were corrected by calibration as described previously [39]. Observed high blood pressure (obsHBP) was categorized for clinical elevation (>130 mmHg/80 mmHg, as per the American Heart Association criteria [40]) at each time point. At baseline, 12.8% of individuals (339 out of 2656) had obsHBP, independent of hypertension diagnosis, and in year 20, 6.4% (n = 171 of 2656) had obsHBP.

Dietary data were collected using the CARDIA diet history method at baseline and year 20 referencing intake for the previous month [41]. The trained CARDIA interview staff conducted interviews, asking the participants open-ended questions regarding intake of 100 food categories that encompassed 1609 separate food items within the past month. For average daily intake calculations (for sucrose, fructose, sodium, and potassium) it was assumed that one month had 30.42 days. Nutrient amounts were calculated with Version 20 of the Food Table from the Nutrition Coordinating Center at the University of Minnesota. Regarding outliers and missing values, the Nutrition Working Group during visit 1 decided to include all out-of-range values in the participants’ records and to not impute any missing data on serving amounts or frequency. Instead of using absolute number of grams for sucrose and fructose, we transformed these values to percent total daily calories by multiplying the number of grams of sucrose/fructose by 4, dividing this number by the total number of daily calories consumed, and multiplying by 100. We included percent calories obtained from sucrose and/or fructose on both year 0 and 20 of follow-up to avoid confounding associated with a single assessment and to account for changes in individuals’ diets as they aged. Fructose quantitation was not completed in year 0, and therefore we only have a measure of percent calories from sucrose for year 0. Both fructose and sucrose quantitation was completed for year 20; thus, we calculated percent calories from sucrose and fructose for year 20 of follow-up.

Psychological stress was assessed using the Centre for Epidemiologic Studies Depression Scale (CES-D), a validated self-report psychometric scale [42]. Items 1–20 are scored on a four-point Likert scale and then summed to obtain a total score that ranges from 0 to 60, whereby a higher score indicates more depressive symptomatology.

Physical activity levels were collected using a validated, self-report questionnaire of leisure time and occupational physical activity [43]. Participants were asked about their participation in moderate- (e.g., walking, golfing, bowling) and vigorous-intensity (e.g., biking, running, racquet sports) aerobic activities over the previous 12 month period.

Our primary, secondary, and tertiary outcomes (CVDs, stroke, and all-cause mortality, respectively) were obtained from the most recent visit at year 35 when the cohort was between the ages of 53 and 65.

### 2.3. Statistical Analysis

For statistical analysis, we used the Statistical Package for the Social Sciences (SPSS) version 28.01.0. Prior to analysis, all continuous variable scales were screened for univariate normality with Kolmogorov–Smirnov test and most measures were found to have issues with skewness and kurtosis. A total of 2656 individuals were included in the analysis. We used numbers and percentages for the participant’s characteristics’ description, and comparisons between males and females were performed with a Pearson χ^2^ test. Race and sex were self-reported. Statistical significance was identified with *p*-values of <0.05. We completed multivariate linear regression analyses for factors associated with VAI in males and females. Collinearity of regression coefficients was ruled out with Pearson correlation and Spearman’s rho. We used Cox proportional hazards regression and corresponding hazard ratios for factors associated with primary, secondary and tertiary outcomes: development of any CVD (fatal and non-fatal), development of stroke (fatal and non-fatal), and death, respectively.

## 3. Results

A total of 2656 patients were included in the final analysis. There were 1159 males (44%) and 1497 females (56%). Participant characteristics are presented in Table 1. The two groups were significantly different for dietary patterns, whereby males had a higher sodium intake both at baseline and at year 20 of follow-up (*p* < 0.001 for both) while females had a higher sucrose intake at year 20 of follow-up (*p* = 0.002). Females had higher depression scores, expressed as Center for Epidemiological Studies Depression (CES-D) *p* < 0.001 at baseline and *p* = 0.021 at year 20 of follow-up, although the absolute differences between CES-D scores were small. Males had lower left and right carotid PWV compared to females (*p* < 0.001 for both), while they had higher carotid IMT (*p* < 0.001). Consequently, VAI was found to be higher in males compared to females (*p* < 0.001). High blood pressure diagnosis was more frequent in males than females at baseline, while the cohort was between 18 and 30 years of age; however, this difference was no longer present 20 years later. At baseline, heart and kidney problems were more frequent in females compared to males (*p* = 0.007 and *p* < 0.001, respectively).

### 3.1. Association of Demographic, Dietary, and Stress Predictors with VAI

Dietary, depression, and demographic predictors of mid-life VAI are shown in Table 2. We completed multivariate linear regression analyses for factors measured at baseline, when the cohort had a median age of 26 years, as well as when predictors were measured 20 years later, which is the same time point at which carotid ultrasounds were obtained and at which we calculated VAI. The predictors included in the multivariable model were dietary sucrose/fructose, sodium, potassium, age, BMI, race, aerobic exercise, and obsHBP (BP > 130/80 mmHg, irrespective of hypertension diagnosis). By considering these predictors at baseline, we evaluated their persistent effects on VAI; meanwhile, through considering them at the time when VAI was obtained we evaluated their acute effects on VAI. For the predictors measured at baseline, age was the only factor significantly associated with increased VAI in both males and females (B-weight = 0.163, *p* < 0.001, and B-weight = 0.154, *p* < 0.001, respectively), while only in females was an association observed for CES-D scores (B-weight 0.063, *p* = 0.015). Some sex-specific patterns were observed for the acute effects of the predictors on VAI. In males, sodium intake was directly correlated (B-weight = 0.145, *p* = 0.003) and potassium intake was inversely correlated with VAI (B-weight = −0.160, *p* < 0.001), while sodium and potassium intake at baseline did not affect VAI in females. Additionally, in males, having had at least 1 h per month of aerobic exercise (run, bike, and/or racquet sports) consistently over the past 12 months was inversely correlated with VAI (B-weight = −0.85, *p* = 0.007). In both males and females, age was directly correlated with VAI (B-weight = 0.171, *p* < 0.001, and B-weight = 0.127, *p* < 0.001).

### 3.2. Predictors of CVDs, Stroke, and Death in the CARDIA Cohort

To determine whether the dietary predictors, depression, and VAI affect cardiovascular morbidity and mortality in the CARDIA cohort, we completed a multivariate Cox proportional hazards analysis shown in Table 3. The latest assessment of outcomes in the CARDIA cohort was in 2021, reporting 178 cases (6.7%) of fatal and non-fatal CVDs, 61 cases (2.3%) of fatal and non-fatal stroke, and 239 (9.0%) of all-cause deaths.

#### 3.2.1. Primary Outcome: CVDs

In Model 1, adjusted for race, sex, age, baseline measures of sucrose intake (expressed as % total calories), CES-D scores, and sodium intake both at baseline and year 20 of follow-up, factors associated with the primary outcome (any CVDs fatal and non-fatal) were BMI of participants recorded at baseline (HR = 1.065, 95% CI [1.039–1.092], *p* < 0.001) and percent calories from fructose recorded at year 20 of follow-up (HR = 1.051, 95% CI [1.009–1.094], *p* = 0.016). Model 2 was additionally adjusted with potassium intake at year 20 of follow-up, and shows the same predictors to be significantly associated with the primary outcome: BMI (HR = 1.067, 95% CI [1.041–1.094], *p* < 0.001) and fructose intake (HR = 1.049, 95% CI [1.006–1.093], *p* < 0.024). In Model 3, we added the covariates of physical activity, both as a self-reported physical activity level as well as specific metrics of consistent aerobic activity. Covariates added to Model 3 included physical activity level, and having had at least 1 h of aerobic activity (run, bike, and/or racquet sports) for 1 h per month consistently for the last 12 months. Even after the adjustment for physical activity, BMI was still a significant predictor of CVDs (HR = 1.041, 95% CI [1.041–1.094], *p* < 0.001), as was dietary fructose (HR = 1.049, 95% CI [1.007–1.093], *p* < 0.022).

#### 3.2.2. Secondary Outcome: Stroke

The BMI of the cohort at baseline was also significantly associated with the secondary outcome, stroke (fatal and non-fatal) (HR = 1.069, 95% CI [1.025–1.115], *p* = 0.002), as was having BP above 130/80 mmHg (obsHBP) at the same timepoint (HR = 2.002, 95% CI [1.078–3.747], *p* = 0.028). Additional covariates in Models 2 and 3 did not change these associations. In Model 2, BMI was associated with HR = 1.068, 95% CI [1.024–1.115], *p* = 0.002 and obsHBP was associated with HR = 2.020, 95% CI [1.083–3.771], *p* = 0.027. In Model 3, the significance of BMI (HR = 1.069, 95% CI [1.024–1.116], *p* = 0.002) and obsHBP (HR = 1.984, 95% CI [1.061–3.711], *p* = 0.032) were maintained.

#### 3.2.3. Tertiary Outcome: Death

Baseline BMI recorded when the cohort was in their young adulthood was the only predictor significantly associated with all-cause mortality more than three decades later in all three models. In Model 1, HR = 1.036, 95% CI [1.011–1.061], *p* = 0.004. In Model 2, HR = 1.036, 95% CI [1.011–1.061], *p* = 0.004. In Model 3, HR = 1.035, 95% CI [1.011–1.060], *p* = 0.004.

## 4. Discussion

We designed this retrospective observational study to assess how dietary behaviors and psychological stress in young adult White and Black males and females impact vascular aging and CVD risk by mid-life. We found the predictors of VAI at mid-life to be sex-specific whereby dietary sodium, potassium, and aerobic activity were significant predictors in males, while depression scores during young adulthood were a significant predictor in females (Figure 1). Moreover, fructose consumption in mid-life was found to be a significant predictor of CVDs 15 years later (Figure 1). Likewise, BP > 130/80 mmHg during young adulthood, irrespective of hypertension diagnosis, was found to be associated with an outcome of stroke 35 years later (Figure 1). Lastly, BMI measured during young adulthood was a significant predictor of CVDs, stroke, and death (Table 3).

Our finding that the predictors of vascular aging at mid-life are sex-specific supports the importance of stratifying data by sex, especially data that include pre-menopausal females. VAI was found to be higher in males compared to females (15.00 ± 2.74 vs. 14.39 ± 2.33, *p* < 0.001, respectively; Table 1). We found that the fructose and sodium intake when the CARDIA cohort was in their adolescent age were not significantly correlated with VAI at mid-life, but that psychological stress was a significant predictor of VAI in females only. Given that depression scores did not differ between males and females (Table 1), this reflects a direct, persistent effect on vascular aging in females only. Similar to other risk factors of vascular aging, psychological stress causes endothelial dysfunction and vascular inflammation [9]. It is of note that significant associations between VAI and depression scores at year 20 were not observed in our analysis, suggesting that chronic psychological stress, rather than acute, has a negative impact on vascular aging in females. It will be important to investigate the persistent associations between psychological stress and VAI in this aging cohort, especially given the transition to menopause that is occurring concurrently at the time of this article. When considering acute predictors of VAI, we found that only in males, dietary sodium was directly correlated with VAI (B-weight = 0.145, *p* = 0.003), while dietary potassium appears to be protective (B-weight = −0.160, *p* < 0.001). Additionally, having at least 1 h of aerobic activity (run, bike, and/or racquet sports) was also negatively correlated with VAI (B-weight = −0.085, *p* = 0.007) in males. This suggests that in mid-life, decreasing dietary sodium, while increasing dietary potassium and aerobic activity for males appears to confer protection against vascular aging. It is tempting to speculate that perhaps at least one of the reasons why we did not observe such associations between sodium, potassium, and aerobic activity with VAI in females is because their contributions are overpowered by estrogen-dependent β-adrenergic-mediated vasodilation [44].

In our multivariable Cox regression analyses, only fructose consumption at year 20 and BMI at baseline predicted CVDs (primary outcome). Even after we adjusted for potassium intake (Model 2) and consistent aerobic activity (Model 3), these associations still persisted (Table 3). A number of epidemiological studies and subsequent meta-analyses have shown that sugar-sweetened beverages increase the risk of hypertension and CVD morbidity and mortality [35,45]. However, few studies have yet explored the relative impact of various types of sugars on CVD risk. To our knowledge, this is the first study to demonstrate in a longitudinal cohort that fructose, not sucrose, predicts CVD risk. Here, we corroborated the preclinical data which demonstrated that dietary fructose contributes to vascular stiffness both in adolescence and adulthood [29,30,33]. In addition to vascular changes, high fructose consumption has been shown to reduce plasma insulin and leptin levels and increase ghrelin concentrations [46], which may contribute to obesity and thus a pro-inflammatory state. While our findings suggest that fructose is an independent predictor of CVDs, it may also contribute to obesity, which is in agreement with our finding that BMI is another significant predictor of CVD morbidity in the CARDIA cohort. In our analysis for the secondary outcome, stroke, we found that both BMI and having BP > 130/80 mmHg in adolescence were significant predictors in all three models. We chose to assess actual measurements of BP, independent of a hypertension diagnosis, because we believe that this is a more clinically relevant scenario contributing to vascular aging. Persistent shear stress caused by elevated BP is known to cause local and functional changes to vascular compliance [47]. This highlights the importance of BP control, and the elevated risk in scenarios when such is not achieved even with medical therapy. Here, we show that BP > 130/80 mmHg in adolescence doubles the risk of having a stroke 35 years later. BMI in adolescence was the only variable in our study to predict all three outcomes: CVDs, stroke, and death. The link between obesity and CVDs has been shown to be multi-faceted and includes interactions between environment, socioeconomic status, genetics, physical activity behaviors, and internal factors [48]. A prolonged state of energy imbalance (greater energy intake compared to expenditure) leads to changes in the functions of adipose tissue, including increased pro-inflammatory adipokine production. The pro-inflammatory state causes atherogenesis and increased endothelial vasomotor tone, both of which can contribute to CVD risk [49].

A limitation of the present study is its retrospective observational nature. Furthermore, dietary practices are derived from Diet History Questionnaires, which are associated with inherent social desirability bias. Compared to other dietary history methods (i.e., 24-h recall or food records), the diet history method used in the CARDIA study may overestimate absolute intake of nutrients, but this should not interfere with analyses that employ a comparison of means among different subgroups within the CARDIA study and only becomes a potential problem when comparing intakes to populations outside of the CARDIA cohort. Every effort was made to keep the diet history questionnaire consistent at all visits. Regarding outliers and missing values, the Nutrition Working Group during visit 1 decided to include all out-of-range values in the participants’ records and to not impute any missing data on serving amounts or frequency.

## 5. Conclusions

In conclusion, our findings suggest that dietary and lifestyle practices that impact vascular aging during adolescence and into mid-life are sex-specific. Lowering psychological stress was found to be important in females, while lowering dietary sodium and increasing potassium and aerobic activity was found to be significantly correlated with vascular aging in males. Furthermore, limiting fructose consumption may confer cardiovascular protection later into mid-life. High fructose consumption may contribute to a higher BMI, which we found to be a significant predictor of CVDs, stroke, and death. Given the aging population in the U.S. and the challenges of large-scale behavior change, there is a need to find small-scale behavioral changes that can help prevent the vascular aging process [10,50]. This study provides evidence that focus on nutrient-specific changes (i.e., lowering fructose in males and females, and focusing on sodium and potassium in males) that are within the purview of overall dietary pattern recommendations may at least partially mitigate vascular aging in early adulthood and the consequential CVD risk in mid-life.

## Figures and Tables

**Figure 1 nutrients-16-00127-f001:**
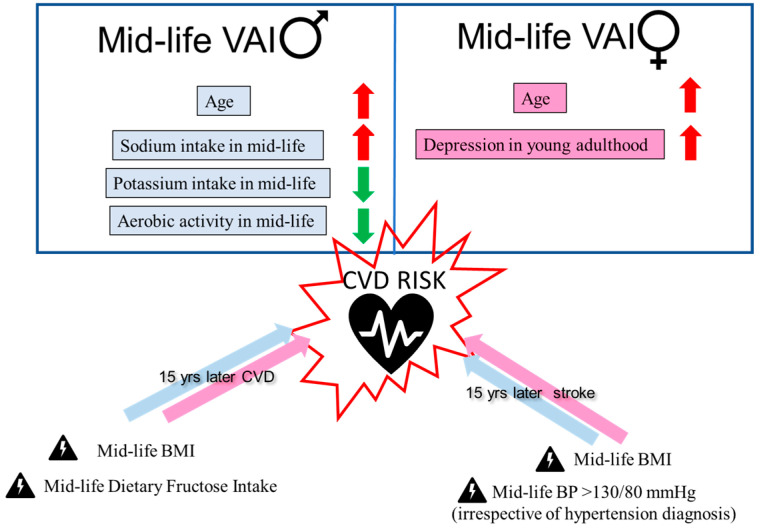
Dietary and depression predictors of VAI and CVD risk. Disparity of dietary and exercise predictors of VAI between males and females is shown. Red arrows indicate increase in VAI and green arrows indicate improvement in VAI. In both males and females, mid-life BMI was associated with increased risk of CVDs and stroke, while dietary fructose intake was specific for CVDs and BP > 130/80 mmHg was specific for stroke.

**Table 1 nutrients-16-00127-t001:** Baseline and year 20 characteristics of the study sample stratified by sex.

	Male (n = 1159)	Female (n = 1497)	*p* Value
**Continuous variables, Mean (SD)**			
Age at baseline	26 ± 3	26 ± 3	0.685
BMI at baseline	23.5 ± 4.6	23.4 ± 4.9	0.906
% calories from sucrose at baseline	6.11 ± 4.82	6.28 ± 6.14	0.337
CES-D baseline	1.42 ± 0.64	1.61 ± 0.80	<0.001
% calories from fructose at year 20	4.38 ± 3.05	4.52 ± 3.65	0.061
% calories from sucrose at year 20	7.07 ± 3.66	7.63 ± 4.11	0.002
Sodium (mg) at year 20	3679 ± 2153	2694 ± 1454	<0.001
Potassium (mg) at year 20	3318 ± 1621	2754 ± 1344	<0.001
CES-D scores at year 20	8.26 ± 6.87	9.35 ± 8.23	0.021
Right carotid PWV (cm/s) at year 20	55.60 ± 15.92	58.90 ± 17.65	<0.001
Left carotid PWV (cm/s) at year 20	52.50 ± 15.65	56.80 ± 17.20	<0.001
cIMT (mm) at year 20	0.62 ± 0.19	0.57 ± 0.16	<0.001
Vascular Aging Index at year 20	15.00 ± 2.74	14.39 ± 2.33	<0.001
**Categorical variables, Numbers (%)**			
HBP observed at baseline	228 ± 8.6	111 ± 4.2	<0.001
HBP observed at year 20	75 ± 6.4	97 ± 6.5	0.993
Baseline High Cholesterol	24 ± 2.1	29 ± 1.9	0.669
Baseline Heart Problems	48 ± 4.1	102 ± 6.8	0.007
Baseline Diabetes	3 ± 0.2	9 ± 0.6	0.216
Baseline Kidney Problems	20 ± 1.7	77 ± 5.2	<0.001
On CVD Medications at Baseline	1 ± 0.08	4 ± 0.2	0.31
Run at least 1 h/month in the past 12 months	122 (10.5)	83 (5.5)	<0.001
Bike at least 1 h/month in the past 12 months	460 (17.4)	394 (14.9)	<0.001
Racket sport at least 1 h/month in the past 12 months	514 (19.4)	465 (17.6)	<0.001

Values are means (SD) for continuous variables and numbers (%) for categorical variables; Comparisons were made with Pearson χ2 test. HBP, high blood pressure, BP > 130/80 mmHg; CES-D, Center for Epidemiological Studies Depression.

**Table 2 nutrients-16-00127-t002:** Dietary, depression, and demographic predictors of mid-life VAI.

	Male			Female		
	b-Weight	B-Weight	*p* Value	b-Weight	B-Weight	*p* Value
**Predictors measured at baseline**						
Age	**0.130**	**0.163**	**<0.001**	**0.100**	**0.154**	**<0.001**
Depression scores	0.067	0.016	0.598	**0.186**	**0.063**	**0.015**
Dietary sodium	0.000	0.037	0.212	0.000	0.004	0.884
% calories from sucrose	0.016	0.027	0.363	0.004	0.012	0.661
BMI	0.019	0.032	0.274	0.005	0.011	0.678
obsHBP	0.345	0.050	0.094	−0.039	−0.004	0.866
**Predictors measured at year 20 of follow-up**						
Age	**0.137**	**0.171**	**<0.001**	**0.079**	**0.127**	**<0.001**
CES-D score	−0.008	−0.021	0.514	0.009	0.033	0.236
% calories from fructose	−0.028	−0.031	0.349	0.026	0.042	0.157
Dietary sodium	**0.000**	**0.145**	**0.003**	0.000	0.023	0.575
Dietary potassium	**0.000**	**−0.160**	**<0.001**	0.000	−0.059	0.150
obsHBP	0.110	0.010	0.757	0.101	0.011	0.694
Aerobic activity (run/bike/racquet sports)	**−0.763**	**−0.085**	**0.007**	−0.400	−0.042	0.137

Unstandardized (b-weight) and standardized (B-weight) regression coefficients, significant predictors are bolded for emphasis; n = 1159 (males), n = 1497 (females). ObsHBP, BP > 130/80 mmHg, irrespective of hypertension diagnosis; CES-D, Center for Epidemiological Studies Depression. Collinearity of regression coefficients was ruled out with Pearson correlation and Spearman’s rho.

**Table 3 nutrients-16-00127-t003:** Multivariate Cox proportional hazards analysis: Primary, secondary, and tertiary outcomes in all patients with regards to dietary, depression, and demographic predictors.

	Model 1	*p*-Value	Model 2	*p*-Value	Model 3	*p*-Value
**Primary Outcome: any CVDs (fatal or non-fatal)**						
**BMI**	**1.065 [1.039–1.092]**	**<0.001**	**1.065 [1.039–1.092]**	**<0.001**	**1.067 [1.041–1.094]**	**<0.001**
obsHBP baseline	1.321 [0.878–1.988]	0.182	1.323 [0.879–1.992]	0.179	1.287 [0.855–1.938]	0.227
obsHBP Y20	1.037 [0.577–1.862]	0.904	1.039 [0.578–1.866]	0.898	1.025 [0.571–1.839]	0.934
**% cal fructose Y20**	**1.051 [1.009–1.094]**	**0.016**	**1.049 [1.006–1.093]**	**0.024**	**1.049 [1.007–1.093]**	**0.022**
% cal sucrose Y20	0.981 [0.942–1.022]	0.369	0.981 [0.942–1.022]	0.358	0.980 [0.941–1.021]	0.339
CES-D Y20	1.004 [0.983–1.025]	0.698	1.004 [0.984–1.026]	0.677	1.003 [0.982–1.024]	0.779
VAI	0.999 [0.941–1.060]	0.968	1.000 [0.942–1.062]	0.998	0.999 [0.941–1.061]	0.985
**Secondary Outcome: stroke (fatal or non-fatal)**						
**BMI**	**1.069 [1.025–1.115]**	**0.002**	**1.068 [1.024–1.115]**	**0.002**	**1.069 [1.024–1.116]**	**0.002**
**obsHBP baseline**	**2.010 [1.078–3.747]**	**0.028**	**2.020 [1.083–3.771]**	**0.027**	**1.984 [1.061–3.711]**	**0.032**
obsHBP Y20	0.382 [0.090–1.610]	0.190	0.389 [0.092–1.645]	0.199	0.385 [0.091–1.627]	0.194
% cal fructose Y20	1.046 [0.975–1.121]	0.210	1.032 [0.959–1.111]	0.399	1.033 [0.960–1.112]	0.380
% cal sucrose Y20	0.947 [0.877–1.021]	0.156	0.946 [0.877–1.020]	0.147	0.944 [0.875–1.018]	0.133
CES-D Y20	0.996 [0.960–1.033]	0.832	0.998 [0.962–1.035]	0.897	0.996 [0.960–1.034]	0.845
VAI	0.857 [0.857–1.068]	0.433	0.9664 [0.863–1.078]	0.521	0.964 [0.861–1.079]	0.526
**Tertiary Outcome: death**						
**BMI**	**1.036 [1.011–1.060]**	**0.004**	**1.036 [1.011–1.061]**	**0.004**	**1.035 [1.011–1.060]**	**0.004**
obsHBP baseline	1.042 [0.707–1.538]	0.834	1.041 [0.705–1.536]	0.841	1.047 [0.709–1.547]	0.818
obsHBP Y20	0.797 [0.450–1.411]	0.436	0.793 [0.447–1.404]	0.425	0.794 [0.448–1.406]	0.429
% cal fructose Y20	1.017 [0.978–1.057]	0.406	1.019 [0.980–1.060]	0.345	1.019 [0.980–1.060]	0.344
% cal sucrose Y20	0.985 [0.951–1.021]	0.411	0.986 [0.952–1.021]	0.432	0.986 [0.952–1.022]	0.441
CES-D Y20	0.997 [0.979–1.016]	0.750	0.997 [0.978–1.015]	0.718	0.997 [0.978–1.015]	0.718
VAI	0.980 [0.928–1.035]	0.466	0.978 [0.926–1.033]	0.433	0.978 [0.926–1.034]	0.436

Significant predictors are bolded for emphasis. Model 1: adjusted for race, sex, % calories from sucrose baseline, depression scores baseline, sodium intake at baseline, and sodium intake at year 20. Model 2: adjusted for race, sex, % calories from sucrose baseline, depression scores baseline, sodium intake at baseline, sodium intake at year 20, and potassium intake at year 20. Model 3: adjusted for race, sex, age, % calories from sucrose baseline, depression scores baseline, sodium intake at baseline, sodium intake at year 20 and potassium intake at year 20, physical activity level, having had at least 1 h of aerobic activity (run, bike, and/or racquet sports) consistently for the last 12 months. n = 2221 (primary outcome), n = 2221 (secondary outcome), n = 2240 (tertiary outcome). ObsHBP = BP > 130/80 mmHg; CES-D = Center for Epidemiological Studies Depression, VAI = vascular aging index.

## Data Availability

Data will be made available upon reasonable request from the corresponding author (D.K.).

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
