# Peer review of "Longitudinal Associations of Dietary Fructose, Sodium, and Potassium and Psychological Stress with Vascular Aging Index and Incident Cardiovascular Disease in the CARDIA Cohort"

_nutrients, 2023, doi:10.3390/nu16010127_

Round 1

Reviewer 1 Report

Comments and Suggestions for Authors

The manuscript entitled ‘Longitudinal Associations of Dietary Fructose and Psychological Stress with Vascular Aging Index and Incident Cardiovascular Disease in the CARDIA Cohort’ written by Meaghan Osborne et al. presents interesting results regarding the impact of diet and lifestyle practices on CVD risk. The study has shown in a large cohort that lowering psychological stress, dietary sodium, and fructose, as well as increasing potassium intake and aerobic activity, could protect against vascular aging and cardiovascular disease. The manuscript is clear, scientifically sound, and presented in a well-structured manner. The conclusions are consistent with the evidence. The English language is correct and clear.

General concept comments

    1.      Table 1 shows differences between males and females included in the study. Many of them were pointed as significant because of statistical significance; however, some of those differences are quite small in values (ex. CES-D baseline 1.42 vs. 1.61) and perhaps their significance should be questioned? Please address this issue in the text.

      2.      The results are clearly presented, but the quality of the manuscript could be increased by adding a simple graphic that presents the relationships between significant predictors and CVD outcomes at the analyzed time points.

Specific comments

I have only a few minor remarks to make the text more clear.

      1.      In the Abstract section, the abbreviation CES-D should be explained.

     2.      In the Materials and Methods section, a definition of a baseline time point could be more highlighted.

     3.      There are also some missing spaces before square brackets with numbers of references (ex. in lines 30, 32, 302, and 317).

    I believe that my remarks help the authors to increase the quality of their work.

Author Response

We thank the reviewer for commendation of the manuscript for clarity and scientific soundness and for providing your valuable feedback in helping improve our manuscript. Please find below point-by-point response to your recommendations:

General concept comments:

  1. Indeed we agree, some of the differences are intuitively small, although statistically significant. We appreciate that you point this out. In the main text, we added on line 208 that the absolute differences in CES-D scores were small irrespective of their statistical significance. We did not however, place too much value on these small differences in the main analysis. Table 1 represents Pearson correlations, and as such, is associated with its limitations.
  2. Thank you for this suggestion, that is a wonderful idea. We now included a graphic that summarizes the findings of the main analysis.

Specific comments:

  1. Thank you, this is now corrected. We deleted CES-D and used depression instead to preserve the 200 word count limit on the Abstract.
  2. We clarified this further by highlighting the definition of baseline on lines 126 and 129 to better drive the point that year 0 = baseline = median age 26 years.
  3. Thank you for catching these, they are now corrected.

Reviewer 2 Report

Comments and Suggestions for Authors

The authors explored extensively the impact of different dietary patterns and psychological status on vascular aging index and cardiovascular disease occurrence. Although, the results are interesting and useful (with potential impact on preventive clinical practice), some issues should be addressed:

1) The following statement: “In the present investigation we included the participants from the CARDIA (Coro-104 nary Artery Risk Development in Young Adults) study” – should be moved to methods section, rather than introduction (as it specifically mentions the cohort of patients analyzed);

2) Also, the following data: “We found that baseline psychological stress was positively associated with vascular aging in females. While in males, sodium intake positively predicted vascular aging and potassium intake inversely predicted vascular aging. Fructose consumption was a significant predictor of CVD risk while having high blood pressure at baseline was a significant predictor of stroke risk” – should be moved to results section, as it highlights the main findings of the article (and not in introduction);

3) The objectives and the novelty of the study should be better highlighted in the introduction section (rather than presenting methodological data or results);

4) The authors should specifically mention in the abstract “year 20 of follow-up”, as it is confounding with participants’ age;

5) The inclusion and inclusions criteria should be described in more detail than in the present form (it is not clear what kind of participants were enrolled, as well as their baseline cardiovascular risk based on inclusion and exclusion criteria). Also, in the methods section it should be mentioned that it was a retrospective observational study;

6) It should be mentioned in the results, that sodium and potassium intake at baseline did not affect VAI in females (as it did in males);

7) It would be interesting to discuss potential explanations for disparities of sodium and potassium intake on outcomes in males and females;

8) The title of manuscript should be probably modified, as the authors investigated not only the impact of dietary fructose, but also of different dietary behaviors.

Comments on the Quality of English Language

Minor English editing is required.

Author Response

We thank the reviewer for finding our results interesting and useful and for taking the time to provide suggestions for improving our manuscript. Please find below the point-by-point responses to your specific suggestions:

1) We left this sentence in the Introduction because we believe it is important to mention the cohort we worked with. We agree that it should be also included (i.e. repeated) in the Methods section, and so we did that (line 123). By naming the cohort used for the study we are able to set up this paragraph such that the novelty of the study is highlighted, as the reviewer pointed out was needed in their comment #2 (please see below).

2) These sentences have been removed from the Introduction, we agree that as summary of results they don’t really belong in the Introduction. Please also note that the rest of the paragraph now highlights the novelty of the study.

3) By completing suggestion # 2 above, we have now better highlighted the objective and the novelty of the study.

4) We did this, thank you for pointing this out.

5) We now clarified the inclusion criteria of the CARDIA study (line 131) as well as our inclusion criteria (line 133). The CARDIA study enrolled healthy young men and women of Black and White race, and we only considered those individuals who attended the carotid Doppler scans at year 20 of follow-up in order to minimize missing data and confounding effects thereafter.

6) This statement was added (line 239), thank you for this point of clarification in highlighting the disparity that exist between males and females.

7) We included the potential explanation for this disparity (line 319-322).

8) Agreed. We modified the title which now includes sodium and potassium as dietary behaviors.